# Towards Robust 3D Pose Transfer with Adversarial Learning

## Abstract

3D pose transfer that aims to transfer the desired pose to a target mesh is one of the most challenging 3D generation tasks. Previous attempts rely on well-defined parametric human models or skeletal joints as driving pose sources, where cumbersome but necessary pre-processing pipelines are inevitable, hindering implementations of the real-time applications. This work is driven by the intuition that the robustness of the model can be enhanced by introducing adversarial samples into the training, leading to a more invulnerable model to the noisy inputs, which even can be further extended to the real-world data like raw point clouds/scans without intermediate processing. Furthermore, we propose a novel 3D pose Masked Autoencoder (3D-PoseMAE), a customized MAE that effectively learns 3D extrinsic presentations (i.e., pose). 3D-PoseMAE facilitates learning from the aspect of extrinsic attributes by simultaneously generating adversarial samples that perturb the model and learning the arbitrary raw noisy poses via a multi-scale masking strategy. Both qualitative and quantitative studies show that the transferred meshes given by our network result in much better quality. Besides, we demonstrate the strong generalizability of our method on various poses, different domains, and even raw scans. Experimental results also show meaningful insights that the intermediate adversarial samples generated in the training can hinder the existing pose transfer models.

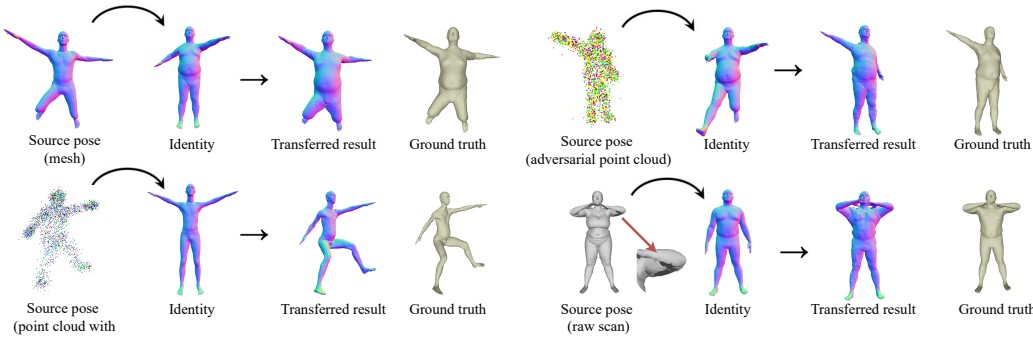

Figure 1: Examples of our 3D pose transfer results on various pose sources, showing strong robustness and generalizability. The pose source includes clean mesh (top left) and point clouds with Gaussian noise (bottom left) from SMPL-NPT dataset (Wang et al., 2020), the adversarial sample of point cloud generated by our method (top right), and raw scan (bottom right) from DFAUST dataset (Bogo et al., 2017). Identity meshes are from the SMPL-NPT dataset (Wang et al., 2020) and the FAUST (Bogo et al., 2014) (bottom right) dataset. Our method can achieve promising pose transfer performance even on the *incomplete* raw scan (bottom right), which is extremely challenging. More experimental results and details can be found in Appendix.

## 1 Introduction

As a promising and challenging task, 3D pose transfer has been consistently drawing research attention from the computer vision community (Song et al., 2021; Wang et al., 2020; Chen et al., 2022; Liao et al., 2022). The task aims at transferring a source pose to a target identity mesh and keeping

the intrinsic attributes (i.e., shape) of the identity mesh. Aside from pure research interests, transferring desired poses to target 3D models has various potential applications in the film industry, games, AR/VR, etc.

To achieve data-driven learning, existing 3D pose transfer methods rely on different prerequisites to the data sources, which severely limits their further real-world implementations. Firstly, many existing 3D pose transfer methods (Ma et al., 2021; Wang et al., 2021b) cannot directly be generalized to unseen target meshes, and training on the target meshes is inevitable for them to learn the priors of the target shape. Secondly, some studies (Groueix et al., 2018; Aberman et al., 2020; Liao et al., 2022) assume that the paired correspondences between the pose and identity meshes are given, such as annotated landmarks/meshes, or T-posed target meshes, which also involves extra manual efforts to obtain. Lastly, all previous attempts of 3D pose transfer rely on pre-processed and clean source poses to drive the target meshes. However, acquiring those clean data is laborious: cumbersome but necessary pre-processing pipelines are inevitable. For instance, to register raw human scans/point clouds to well-defined parametric human models (e.g., SMPL series (Loper et al., 2015; Bogo et al., 2016; Pavlakos et al., 2019)), it will take roughly 1-2 minutes to process merely a single frame (Wang et al., 2021a), hindering implementations of the real-time applications.

Inspired by the scaling successes of adversarial learning in the computer vision community (Goodfellow et al., 2014; Carlini et al., 2019; Croce & Hein, 2020; Awais et al., 2021; Huang et al., 2021) for enhancing the robustness of the models, we experiment with applying adversarial training to 3D pose transfer tasks. As shown in Fig. 1, we wish to utilize the strength of adversarial learning to enhance the robustness and generalizability of the model, and go beyond so that conducting pose transfer on unseen domains or even directly from raw scans can be possible.

However, although the above idea is intuitive, *it's not feasible to naively extend existing adversarial training algorithms (Zhang et al., 2021; Xiang et al., 2019; Tsai et al., 2020) to the 3D pose transfer task.* Primarily, the current methods (Zhang et al., 2021; Wu et al., 2020) generate adversarial samples based on discriminative adversarial functions, and to our knowledge, there is no adversarial samples proposed specifically for the 3D generative tasks (i.e., how to justify if an adversarial sample is good based on generated results). Moreover, previous approaches using 3D adversarial samples, such as (Wu et al., 2020; Zhang et al., 2021), do so by taking them as pre-computed input data and leaving them untouched for the entire training procedure. This protocol is not practical for our task, as models need to learn the latent pose space via gradients. Thus, we proposed a novel adversarial learning framework with a new adversarial function and on-the-fly computation of adversarial samples, which enables the successful application of adversarial training of the generative models.

Another novel ingredient in this paper is inspired by the recent powerful capability of masked autoencoding (MAE) architectures (He et al., 2022) in the computer vision community. We adopt the idea of MAE to 3D pose transfer task by implementing a new model, called 3D-PoseMAE, that empathizes the learning of extrinsic presentations (i.e., pose). Specifically, unlike any existing 3D MAE-based models (Yu et al., 2021; Zhang et al., 2022; Pang et al., 2022) that merely put efforts into depicting spatial local regions to capture the geometric and semantic dependence, 3D-PoseMAE exploits a multi-scale masking strategy to aggregate consistent sampling regions across scales for robust learning of extrinsic attributes. Besides, we observe that local geometric details (wrinkles, small tissues, etc.) from the pose sources are intuitively unessential for pose learning. Thus, we argue that traditional 3D spatial-wise attention/correlation operation (Chen et al., 2022; Song et al., 2021) might involve a lot of redundant geometric information for pose learning. Instead, we adopt a progressive channel-wise attention operation to the 3D-PoseMAE so that the attention is operated gradually and fully on the latent pose code, making the presentations more compact and computationally efficient.

The contributions are summarized as follows:

- We work on the robustness problem of 3D pose transfer. To our knowledge, it is the first attempt made to approach 3D pose transfer from the aspect of adversarial learning. Besides, the current community holds the assumption that preprocessed pose sources are available. Instead, we break this limitation by providing a new research entry that generates adversarial samples to simulate noisy inputs and even raw scans so that conducting pose transfer directly on raw scans and point clouds is made possible.

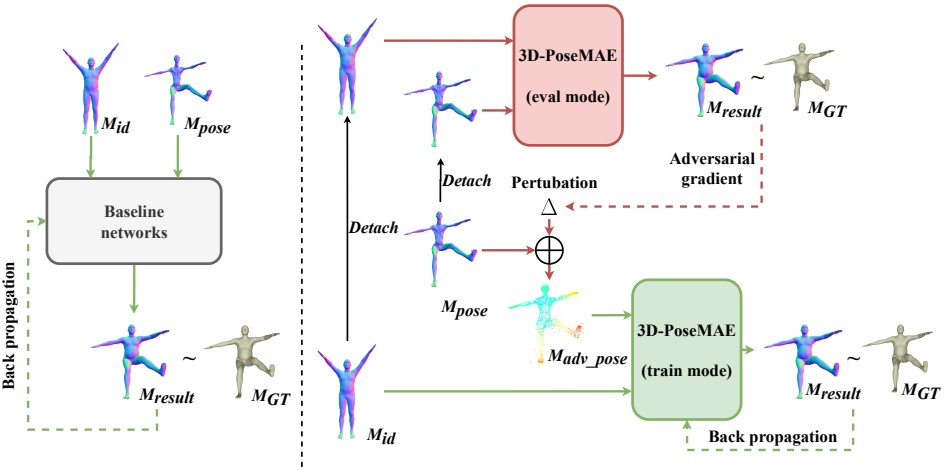

Figure 2: **Left:** The traditional pipeline used in previous methods (Wang et al., 2020; Song et al., 2021; Chen et al., 2022; Liao et al., 2022) for 3D pose transfer. The model is trained with clean mesh inputs without considering the robustness of the noisy inputs. We use the symbol ∼ to generally refer to the loss term, which differs according to the actual condition. **Right:** Our method. Our method utilizes the strength of adversarial learning to enhance the robustness and generalizability of the model. It consists of an adversarial sample generating flow(top part in red) and a pose transferring flow(bottom part in green). The two flows happen iteratively during the adversarial training and the adversarial samples are calculated on-the-fly. Note that $M_{id}$, $M_{pose}$, $M_{result}$, and $M_{GT}$ stand for the identity, pose, generated meshes, and ground truths, the same as below.

- We introduce a novel adversarial learning framework customized for the 3D pose transfer task with a novel adversarial function and on-the-fly computation of adversarial samples in backpropagation. It's the first time that on-the-fly computation of adversarial samples appears in a 3D generative deep learning pipeline.
- We propose the 3D-PoseMAE, an MAE-based architecture for 3D pose transfer with carefully designed components to capture the extrinsic attributions. It exploits a multi-scale masking strategy to learn the global features and a progressive channel-wise attention operation which is more compact than the traditional spatial ones. The 3D-PoseMAE shows encouraging performances in both computational efficiency and generative ability.
- Quantitative or qualitative experimental results on various different datasets and different data sources show that our proposed method achieves promising performances with substantial robustness to noisy inputs and the generalizability to noisy raw scans from the real world without any manual intervention. Code will be made available.

## 2 METHODOLOGY

We first present a general introduction to the whole pipeline of the proposed method as shown in Fig. 2. Compared to the traditional 3D pose transfer pipeline, we introduce adversarial learning. As shown in Fig. 2, our method consists of an adversarial sample generation procedure (top part in red flow) and a pose transferring (bottom part in green flow) procedure. In the top adversarial sample generation flow, the proposed 3D-PoseMAE model will be set as *eval* mode for obtaining the gradient of the data. By using the gradient of the output of 3D-PoseMAE, we can obtain the perturbation to the meshes, resulting in adversarial samples. In the bottom pose transferring flow, the 3D-PoseMAE model works the same as traditional 3D pose transfer models in a *train* mode, but the pose inputs are replaced with adversarial samples, leading to the adversarial training. Note that the inputs $M_{pose}$ and $M_{id}$ of 3D-PoseMAE need to be detached to ensure the backpropagation of the two flows works without interference.

Next, the overall architecture of the 3D-PoseMAE network together with each component will be introduced, followed by the implementation details of adversarial training.

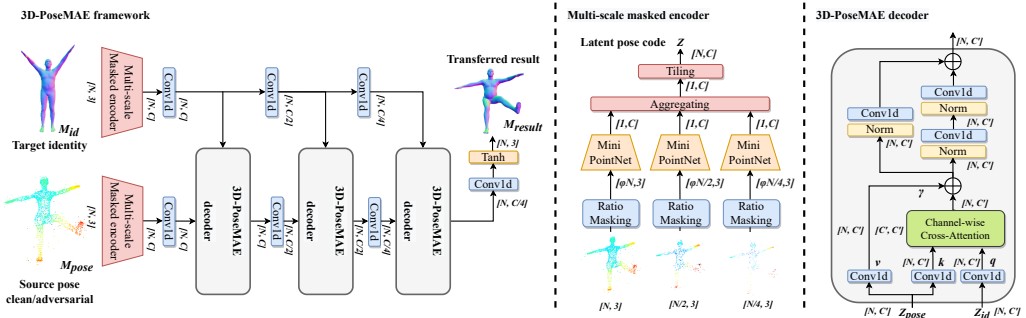

Figure 3: An overlook of our 3D-PoseMAE. The left part is the whole architecture of the 3D-PoseMAE. The middle and right parts illustrate the architectural details of one multi-scale masked encoder and one 3D-PoseMAE decoder, respectively. The 3D-PoseMAE borrows the idea from the work of (He et al., 2022) but is extensively extended to 3D data processing and especially for the 3D pose transfer task. Note that $Z$ stands for the encoded pose feature. $Z_{pose}$ and $Z_{id}$ stand for the specific encoded pose features from pose and identity. Subscripts are the dimensional shape of variables.

## 2.1 3D-POSEMAE

An overview of the 3D-PoseMAE network is presented in Fig. 3. Similar to previous works (Wang et al., 2020; Song et al., 2021; Chen et al., 2022), 3D-PoseMAE takes a source pose (in the form of a mesh or a point could) and a target mesh as input, and outputs a corresponding pose transferred result. In the following, we will introduce each component of the 3D-PoseMAE in an order following the data stream.

**Multi-scale Masked 3D Encoder.** Analogy to the original MAE (He et al., 2022), existing 3D MAE-based architectures (Zhang et al., 2022; Pang et al., 2022; Yu et al., 2021) invariably chose to split a given point cloud into multiple subsets to conduct the masking. However, this protocol involves feature extraction on each subset for later attention and aggregation, which is computationally demanding and only works on small-scale meshes/point clouds (up to 8,192 points). It cannot be extended to tasks with large-size meshes, such as 3D pose transfer, where the size of the target mesh can be more than 27,000 vertices. Thus, we propose a novel multi-scale masking strategy that takes advantage of the masking strategy to boost extrinsic attribute learning. Specifically, as shown in Fig. 3, we regard the input point cloud $M_{pose}$ as the 1-th scale. For the $i$-th scale, $1 \leq i \leq 3$, we randomly downsample the points by $2^{(i-1)}$ times, resulting in a group of the $S$-scale representations of the source pose. Then, we adopt masking at a ratio of $\phi$ to the point cloud at each scale, so that model will be pushed to learn the same extrinsic attribute shared from those scaled representations via Mini-pointNet (Qi et al., 2017b). Next, we aggregate the resulting latent pose code from all scales and form the embedding vector $Z$ with dimension $C$. At last, we tail the size of $Z$ from $[1, C]$ to $[N, C]$ for better integration with the identity mesh (N is the vertex number of the identity mesh, which is flexible according to given target meshes) and feed it into the following 3D-PoseMAE decoders.

**3D-PoseMAE Decoder with Channel-wise Attention.** After obtaining the latent codes of pose and identity $Z_{pose}$ and $Z_{id}$, we need to adopt the Transformer backbone to integrate two latent codes and conduct the decoding and generating. As there are many existing 3D transformer backbones (Zhao et al., 2021; Pang et al., 2022; Chen et al., 2022) and the work of GC-Transformer (Chen et al., 2022) is proposed specifically for 3D pose transfer, we implement our 3D-PoseMAE decoder based on GC-Transformer architecture as shown in the right part of Fig. 3. However, our 3D-PoseMAE decoder has a core difference from any existing 3D Transformer, which is the design of the attention operation. To learn the correlations of given meshes, previous models conduct the attention operation over the spatial channel to perceive the geometric information from the two meshes. Our intuition is that many redundant local geometric representations (wrinkles, small tissues) from the source pose are not essential for pose learning. In other words, we only need the compact pose representations from source poses, it would be encouraging to conduct channel-wise attention where the integration will be fully achieved on the compact pose spaces.

With the above observation, we propose a channel-wise cross-attention module in our 3D-PoseMAE decoder. Firstly, we construct the channel-wise attention map $A$ between $\mathbf{q}$ and $\mathbf{k}$ with the following formula (the subscripts indicate the matrix size):

$$\mathbf{A}_{C' \times C'} = \text{Softmax}(\mathbf{q}_{C' \times N}^T \mathbf{k}_{N \times C'}), \tag{1}$$

where the representations $\mathbf{qk}$ are generated from embedding vectors via different 1D convolution layers (the same as $\mathbf{v}$ below), with the same size as $[N, C']$ ($N$ for vertex number and $C'$ for the channel dimension in the current decoder). Thus, the size of the resulting attention map becomes $[C', C']$. Next, we can obtain the refined latent embedding $Z'_{pose}$ with following

$$Z'_{pose} = \gamma \mathbf{A}_{C' \times C'} \mathbf{v}_{C' \times N}^T + Z_{pose}, \tag{2}$$

where $\gamma$ is a learnable parameter. The remaining design of the 3D-PoseMAE decoder is consistent with the GC-Transformer (Chen et al., 2022), and the whole network structure is presented in Fig. 3. Please refer to the Appendix for more details on network design and parameter setting.

As we can see, by implementing such a channel-wise attention operation, the network can enjoy two major benefits. Firstly, the size of the intermediate attention map in the network changes from $[N, N]$ to $[C', C']$. Thus, the model size will be substantially reduced. Especially when the processing mesh size is huge, e.g., vertex number $N$ could be up to 27,000 while channel size $C'$ is fixed no more than 1,024 in practice. Secondly, the integration of the pose and target information will be more compact with the cross-attention conducted fully channel-wise, avoiding touching the redundant spatial information brought from the source pose. Note that, the fine-grained spatial geometric information of the target meshes is preserved by the gradual integration in the 3D-PoseMAE, which is refined by the compact pose representations from the channel-wise attention mechanism.

**Optimization**. To train the 3D-PoseMAE, we define the full objective function as below:

$$\mathcal{L}_{full} = \mathcal{L}_{rec} + \lambda_{edge} \mathcal{L}_{edge}, \tag{3}$$

where $\mathcal{L}_{rec}$ and $\mathcal{L}_{edge}$ are the two losses used as our full optimization objective, as reconstruction loss and edge loss. $\lambda_{edge}$ is the corresponding weight of edge loss. In Eq. 3, reconstruction loss $\mathcal{L}_{rec}$ is the point-wise L2 norm, and the edge loss $\mathcal{L}_{edge}$ (Groueix et al., 2018) is an edge-wise regularization that prevents flying points with long edges. Note that to make a fair comparison, we only exploit these two loss terms that are used in existing baselines (Wang et al., 2020; Song et al., 2021) and did not apply extra loss terms such as Laplacian loss (Sorkine et al., 2004), central geodesic contrastive (CGC) loss (Chen et al., 2022), although they could boost the generating performance with higher quality meshes and the potential of the losses could depend on the architecture.

## 2.2 ADVERSARIAL TRAINING

An overview of our adversarial learning-based pipeline is presented on the right side of Fig. 2. Below, we introduce the motivation and problem definition of the adversarial training in the pose transfer task, followed by the implementing details.

**Motivation.** It's proven in various computer vision tasks (Carlini et al., 2019; Awais et al., 2021; Huang et al., 2021; Zhu et al., 2021; Wang et al., 2022) that introducing adversarial samples to perturb the neural networks during the training can enhance networks' robustness and increase their generalizability and adaptability to other/unseen domains. It's intuitive to think of applying a similar scheme to the 3D pose transfer task so that transferring poses from unseen domains or even directly from raw scans can be made possible. However, this would not work naively by transferring the existing 3D adversarial sample generating methods to our task due to several issues. Below, we will discuss each issue and illustrate how we approach to the solution.

**Problem Definition.** We define the problem of adversarial training in the 3D pose transfer task as follows. Given a source pose $M_{pose}$ and an identity mesh $M_{id}$, let $F$ be the target pose transfer model (e.g., 3D-PoseMAE) with parameters $\theta$. Then $F(M_{pose}, M_{id}; \theta)$ is the pose transferred mesh generated by the model. For an ideal model we should get: $F(M_{pose}, M_{id}; \theta) = M_{GT}$, with $M_{GT}$ as the ground truth mesh. Then, the problem is converted to an inequality problem: adversarial sample generating method $f$ needs to generate an adversarial sample $M_{adv\_pose} = f(M_{pose})$ that can satisfy:

$$F(M_{adv\_pose}, M_{id}; \theta) \neq M_{GT}. \tag{4}$$

To solve the inequality problem analogy to Eq. 7, existing 3D adversarial sample generating methods for 3D object classification tasks (Zhang et al., 2021; Tsai et al., 2020; Dong et al., 2020; Zheng et al., 2019; Xiang et al., 2019; Kim et al., 2021) convert the inequality into a minimal optimization problem with adversarial loss term $||\mathrm{argmax}_c F(x_{adv}; \theta)_c - t||$ by forcing the prediction to an adversarial sample $x_{adv}$ close to a target class $t$ which is different than the correct one, resulting a successful adversarial sample. This is known as targeted adversarial samples. However, this discriminative-based adversarial function cannot directly be applied to generative tasks (e.g., 3D pose transfer) as a continuous latent space is required to present the pose code.

**Untargeted Adversarial Samples for Pose Transfer.** Some non-targeted adversarial sample generating methods (Moosavi-Dezfooli et al., 2017; Su et al., 2019) convert the above adversarial loss term for targeted adversarial samples into a *reversed* form of $||\mathrm{argmax}_c F(x_{adv}; \theta)_c - t||$ as a minimal optimization problem by pushing away the prediction from the correct class. Inspired by this, we construct an adversarial function similar to our 3D generative task in an intuitive way:

$$f_{adv} = ||F(M_{adv\_pose}, M_{id}; \theta) - M_{GT}||^{-1}. \tag{5}$$

By minimizing the above term, we can push the generated results from the model away from the ground truth mesh, resulting in an adversarial training effect.

**The Magnitude of Adversarial Samples.** To guarantee the adversarial sample is visually similar to the clean data, adversarial sample generating methods (Zhang et al., 2021; Tsai et al., 2020; Dong et al., 2020; Zheng et al., 2019; Xiang et al., 2019; Kim et al., 2021) deploy norms (C&W based adversarial samples) or predefined threshold budget (PGD-based methods) to restrict the perturbations small enough. While in our case, we wish the perturbation to be strong enough so that model can handle it without the need to consider the invisibility issue of the perturbation since the goal is not to obtain effective adversarial samples but a robust model. But if the magnitude of the adversarial sample is too large, the sample can be easily defended by simple pre-processing such as statistical outlier removal (SOR) (Wu et al., 2020), thus failing to contribute to the adversarial training. Thus, to make it closer to real-world applications, we add SOR as pre-processing to all the adversarial samples to filter out those easy samples.

**Adversarial Training for Robustness.** After confirming the adversarial loss function as Eq. 8, we can merge the adversarial samples into the 3D pose transfer training. As mentioned, C&W-based methods can provide better adversarial samples with less visibility. But they all suffer from the time-consuming issue due to the binary search and heavy optimization iteration. According to our preliminary implementation, merging C&W-based methods (Xiang et al., 2019) (the perturbations with original parameter setting) into the pose transfer learning will extend the training time by more than 900 times, which is not feasible. Thus, we deploy PGD-based method (Dong et al., 2020), including the several variants of fast gradient method (FGM) method and PGD method, into the framework for the adversarial training. We encourage readers to refer to the Appendix and (Goodfellow et al., 2014; Dong et al., 2020) for a detailed explanation and implementation of adversarial samples/attacks on 3D data. The training pipeline is demonstrated in Algorithm 1 in the Appendix by taking the FGM method as an example. With this pipeline, we achieve on-the-fly computation of adversarial samples, which enables the generative model to cover the whole latent pose space via gradients. It's worth emphasizing that our method generates adversarial samples directly based on the gradients of arbitrary given meshes, with no need of utilizing SMPL models in the training.

## 3 EXPERIMENTS

In this section, we present comprehensive experiments conducted to evaluate our approach by comparing other state-of-the-art models. Firstly, quantitative evaluations of 3D-PoseMAE are represented with two protocols for both clean samples training and adversarial training. One step further, we qualitatively visualize the strong generalization ability of our method as well as the intermediate-generated adversarial samples. Lastly, we perform ablation studies to evaluate the effectiveness of our methods.

**Datasets.** (1) SMPL-NPT (Wang et al., 2020) is a synthesized dataset containing 24,000 body meshes generated via the SMPL model (Bogo et al., 2016) by random sampling in the parameter space. 16 different identities paired with 400 different poses are provided for training. At the testing

Table 1: Comparison with other methods on SMPL-NPT dataset with training on clean samples.

| Methods | PMD $\downarrow$ ($\times 10^{-4}$) | |
|---|---|---|
| | Seen | Unseen |
| DT (Groueix et al., 2018) | 7.3 | 7.2 |
| NPT-MP (Wang et al., 2020) | 2.1 | 12.7 |
| NPT (Wang et al., 2020) | 1.1 | 9.3 |
| 3D-CoreNet (Song et al., 2021) | 0.8 | - |
| GC-Transformer (Chen et al., 2022) | **0.6** | 4.0 |
| 3D-PoseMAE (Ours) | **0.6** | **3.4** |

Table 2: Comparison with other methods on SMPL-NPT dataset evaluated with adversarial samples.

| Method | PMD $\downarrow$ ($\times 10^{-4}$) | |
|---|---|---|
| | Clean Training | Adversarial Training |
| NPT-MP (Wang et al., 2020) | 307.1 | 62.3 |
| NPT (Wang et al., 2020) | 237.6 | 59.0 |
| GC-Transformer (Chen et al., 2022) | 105.2 | 19.3 |
| 3D-PoseMAE (Ours) | **77.6** | **16.9** |

stage, 14 new identities are paired with those 400 poses used in the training set as the "seen" protocol and 200 new poses as "unseen" protocols. The models are only trained on SMPL-NPT dataset and will be generalized to other datasets. (2) FAUST (Bogo et al., 2014) is a well-known 3D human body scan dataset that the mesh structure of FAUST registrations fits the SMPL body model with 6,890 vertices. We use it for qualitative evaluation. (3) DFAUST (Bogo et al., 2017) dataset is a large human motion sequence dataset that captures the 4D motion of 10 human subjects performing 14 different body motions. We use it for both qualitative and quantitative evaluation.

**Implementation Details.** Our algorithm is implemented in PyTorch (Paszke et al., 2019). All the experiments are carried out on a server with four Nvidia Volta V100 GPUs with 32 GB of memory and Intel Xeon processors. We train our networks for 400 epochs with a learning rate of 0.00005 and Adam optimizer (Kingma & Ba, 2015). The batch size is fixed as 4 for all settings. It takes around 15 hours for pure training (clean samples) on 3D-PoseMAE and 40 hours for adversarial-based training (with FGM, which is the fastest) on 3D-PoseMAE. Please refer to the Appendix for more details on parameter settings and other methods.

The adversarial training of the whole framework is to seek a balance between the magnitude of the adversarial samples and the generative model's robustness. Although it can be controlled by adjusting hyper-parameters such as the adversarial sample budget $\epsilon$, it's intuitive to assume that introducing adversarial samples into the early stage of the training would not contribute as the transferring results are still degenerated and easily fall into local minima and numerical errors in gradient computation. Thus, similar to many existing methods (Cosmo et al., 2020; Chen et al., 2021) that have trade-offs in training, we conduct the training with two stages. For the first 200 epochs, only pure training with clean samples is conducted to stabilize the model and avoid local minima, where the reconstruction loss and edge loss are used. After 200 epochs, the adversarial training starts with adversarial samples added.

## 3.1 QUANTITATIVE EVALUATION

**Evaluation on Clean Training.** We start the evaluation with training models on clean samples to solely verify the performance of 3D-PoseMAE. We follow the classical evaluation protocol for 3D pose transferring from (Wang et al., 2020) to train the model on the SMPL-NPT dataset and evaluate the resulting model with Point-wise Mesh Euclidean Distance (PMD) as the evaluation metric:

$$PMD = \frac{1}{|V|} \sum_{\mathbf{v}} \|M_{\mathbf{result}} - M_{\mathbf{GT}}\|_2^2 . \tag{6}$$

where $M_{\mathbf{result}}$ and $M_{\mathbf{GT}}$ are the point pairs from the ground truth mesh $M_{\mathbf{GT}}$ and generated one $M_{\mathbf{result}}$. The corresponding experimental results are presented in Table 1. As we can see, our 3D-PoseMAE achieves the SOTA performance with the lowest PMD ($\times 10^{-4}$) of: 0.6 and 3.4 on "seen" and "unseen" settings. We denote PMD ($\times 10^{-4}$) as PMD for simplicity in the following. Specifically, although the compared method GC-Transformer has the same performance as ours on the "seen" setting, their model size and runtime (see below) are much larger than ours.

**Evaluation on Adversarial Training.** We next conduct the experiments by evaluating models on adversarial samples to verify the performances of models from the robustness aspect. We continually

Table 3: Evaluation across datasets. Models are trained on SMPL-NPT in an adversarial manner.

| Datasets | Domains | NPT | 3D-PoseMAE (Ours) |
|---|---|---|---|
| DFAUST (Bogo et al., 2017) | Raw scan | 25.21 | **13.70** |
| SMPL-NPT (Wang et al., 2020) | Gaussian noise | 12.66 | **6.13** |

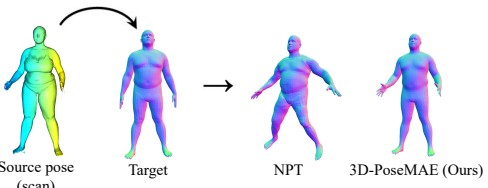

Figure 4: The performance of our method and compared method (Wang et al., 2020) on an unseen raw scan from the DFAUST dataset (Bogo et al., 2017). We can see that the compared method failed to handle the raw scan as a source pose, leading to an arbitrary-generated pose while our method can preserve the original pose in a better visual effect.

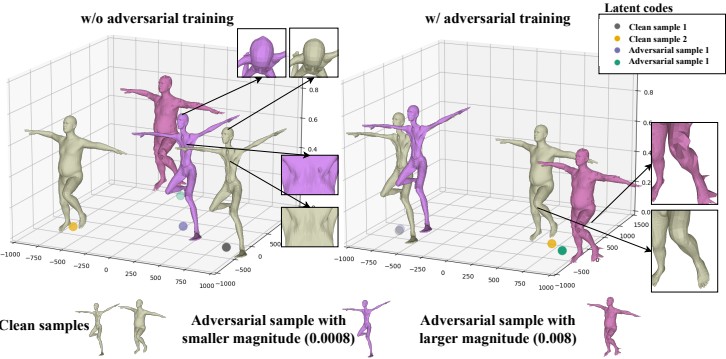

Figure 5: Visualization of latent pose space and the corresponding pose. We present two adversarial samples to qualitatively analyze the patterns of the perturbation.

use the PMD as the evaluation metric. Specifically, to make a fair comparison, we also conducted the adversarial training of two baseline methods, NPT (Wang et al., 2020) and GC-Transformer (Chen et al., 2022), with exactly the same setting for the adversarial training pipeline. To build the testing set of adversarial samples, we directly utilize the "seen" testing set from the SMPL-NPT dataset and generate adversarial samples (using the same attacking strategy, i.e., the FGM) to attack the victim models. The experimental results are presented in Table 2, where we report the result of using the FGM method with an budget of 0.08. More experiments with different types can be found in Appendix. When it comes to the pose transferring with adversarial samples, all the methods' performances will suffer from the degeneration problem compared to using the clean pose as the source. Especially for models without adversarial training, the pose transferring performance will drop dramatically, proving the vulnerability of those "clean" models. Instead, the robustness of all the compared methods has been enhanced considerably with the adversarial training. It is worth noting that our 3D-PoseMAE achieves state-of-the-art performance within both of the training strategies, demonstrating it as a strong baseline network.

**Evaluation on Noisy Inputs and Raw Scans.** We present extra evaluations on raw scans from the DFAUST dataset and meshes with Gaussian noises on the SMPL-NPT dataset in Table 3. Our 3D-PoseMAE outperforms the compared methods by a large margin.

**Runtime & Model Size.** We present Table 4 to demonstrate the computational attribute of 3D-PoseMAE. The runtime is obtained by taking the average inference times in the same experimental settings. As shown in the table, the 3D-CoreNet method (Song et al., 2021) takes the longest time and largest size compared to other deep learning-based methods. The NPT method (Wang et al., 2020) has the shortest inference time as there is no correlation module involved thus, the generation performance is degraded. 3D-PoseMAE achieves notable improvements, while the inference time is also encouraging.

Table 4: Model size and runtime of different methods.

| Model | Correlation module | Model size | Pose Source | Runtime |
|---|---|---|---|---|
| NPT (Wang et al., 2020) | - | 24.2M | Mesh | 0.0044s |
| 3D-CoreNet (Song et al., 2021) | Correlation matrix | 93.4M | Mesh | 0.0255s |
| GC-Transformer (Chen et al., 2022) | Spatial-attention | 48.1M | Mesh | 0.0056s |
| 3D-PoseMAE (Ours) | Channel-attention | 40.7M | Mesh/Raw scan | 0.0048s |

Table 5: Ablation study by progressively enabling each component. The rightmost is from the full 3D-PoseMAE.

| Component | Vanilla | +Multi-scale masking | +Channel Attention |
|---|---|---|---|
| PMD $\downarrow$ ($\times 10^{-4}$) | 0.64 | 0.63 | 0.60 |

Table 6: Difference masking ratio in 3D-PoseMAE.

| Ratio | 0.0 | 0.3 | 0.5 | 0.7 | 0.9 |
|---|---|---|---|---|---|
| PMD $\downarrow$ ($\times 10^{-4}$) | 4.0 | 3.8 | 3.8 | 3.9 | 5.3 |

## 3.2 QUALITATIVE EVALUATION

**Robustness to Various Pose Sources.** Our final goal of introducing adversarial training is to enhance the robustness of the model so that it can be generalized to various noisy and complicated situations. Thus, we qualitatively display the generality of 3D-PoseMAE over various pose sources, as shown in Fig. 1. Besides, we compare our method with the existing pose transfer method (Wang et al., 2020) in Fig. 8, to directly transfer pose from raw scans, and the encouraging performance on raw scans proves that adversarial samples can effectively improve the robustness of the models.

**Visualization of the Perturbations.** We further visualized the perturbations of the generated adversarial samples in Fig. 5. Intriguingly, we find that the majority of generated perturbation happened to locate on the body parts that are consistent with what we usually think of as the key kinetic positions, such as the knees, elbows, and feet.

## 3.3 ABLATION STUDY

**Clean vs. Adversarial Training.** We verify the efficacy of adversarial training against the adversarial samples/noisy inputs in Table 2. We can see that, when being attacked by adversarial samples, all the existing methods enjoy benefits from adversarial training compared to the ones being trained with clean samples, e.g., 19.3 vs. 105.2 for GC-Transformer. The results also prove that *generated adversarial samples can successfully attack the pure models*, leading to degenerated results, e.g., the performance of the NPT model degenerates dramatically with PMD from 1.1 (clean pose source) to 237.6 (attacking pose source).

**Each Component.** We verify the contributions made from each component in the 3D-PoseMAE in Table 5. We disable all the key components as a Vanilla model and enable each step by step. Compared to the Vanilla model, the multi-scale masking strategy, channel-wise cross-attention can consistently improve the generative performance of the task.

**Masking Ratio Scheme.** Lastly, we verify different masking ratios of the multi-scale masking encoder, as shown in Table 6 where we choose the masking ratio as 0.5 for the best performance. The reported results are on the "unseen" setting of the SMPL-NPT dataset.

## 4 CONCLUSION

We work on the robustness problem of the 3D pose transfer, especially on unseen domains and raw noisy inputs from the aspect of adversarial learning. We propose the 3D-PoseMAE with two novel components: a multi-scale masking strategy and a progressive channel-wise attention operation. We further propose a novel adversarial learning framework customized for 3D pose transfer with a new adversarial function and on-the-fly computation of adversarial samples implementation. Experimental results on different datasets and various data sources show that our method achieves promising performances with substantial robustness to noisy inputs. We show that our framework can be successfully extended to noisy raw scans from the real world without any manual intervention.

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

## A  RELATED WORK

**Data-driven 3D Pose Transfer.** Data-driven 3D pose transfer aims to transfer given source poses to target shapes by learning the correspondence between the pose and shape automatically. On the one hand, parametric human model-based methods could bring impressive generated results (Ma et al., 2021; Wang et al., 2021b), but their superb performances largely rely on the need for the priors of the target shape, which limits their generalization ability to unseen target meshes. Besides, registering raw human scans to well-defined parametric human models is also cumbersome and time-consuming (Wang et al., 2021a). On the other hand, some linear blending skinning (LBS) based works (Aberman et al., 2020; Liao et al., 2022) can be extended to unseen target meshes, but they assume that annotated landmarks/mesh points or T-posed target meshes are given, which is also a strong condition. Lastly, to our knowledge, all existing methods are within a strong prerequisite that pre-processed and clean source poses are available to drive the target meshes. However, to our knowledge, no effort has been made to study the robustness of the 3D pose transfer to the noisy inputs and even raw scans.

**3D Adversarial Learning.** Adversarial attacks (Carlini et al., 2019; Croce & Hein, 2020) have drawn considerable research attention in the 2D vision models as their severe threat to real-world deployments since it was proposed (Goodfellow et al., 2014). When it comes to the 3D field, the attack-generating algorithms could be roughly sorted into Carlini& Wagner (C&W) (Carlini & Wagner, 2017), and Projected Gradient Descent (PGD) (Goodfellow et al., 2014) groups. C&W-based attacks (Xiang et al., 2019; Tsai et al., 2020; Zhang et al., 2021) switch the min-max trade-off problem of adversarial training into jointly minimizing the perturbation magnitude and adversarial loss of attacks. But C&W attacks all suffer from time-consuming issues due to the binary search and

optimization iteration. Meanwhile, PGD attacks (Liu et al., 2019; Dong et al., 2020) set the perturbation magnitude as a fixed constraint in the optimization procedure, which can achieve the attack in a much shorter time. But the attack form is limited to point-shifting, unlike C&W attacks that can perform adding or dropping operations on the point clouds. However, there is no strategy proposed specifically for attacking the 3D generative tasks, including pose transfer (i.e., it is unaddressed how to define a successful attack to a generated point cloud/mesh). Besides, research efforts (Wu et al., 2020; Sun et al., 2022; Li et al., 2022) have been made to defend against those attacks on point cloud data for various tasks. But those approaches conduct the defense training directly on 3D precomputed adversarial samples. To our knowledge, there is no existing 3D deep learning pipeline that jointly generates adversarial samples and learns the defense of generative models yet. Note that, for the convenience to reader better understand our work, we avoid calling those methods as attacking methods. Instead, we implement them as adversarial sample-generating methods and use them for the adversarial training process.

**Deep Learning Models on Point Cloud.** The pioneering and representative deep learning models on point cloud include PointNet, PointNet++, and Dynamic Graph CNN (DGCNN) (Qi et al., 2017a;b) as being widely used as benchmark models on various 3D tasks. In the past two years, with the trend of Transformer-based and MAE-based architectures (He et al., 2022) in the computer vision community, many 3D-variant models (Zhao et al., 2021; Yu et al., 2021; Zhang et al., 2022; Pang et al., 2022) have been proposed. An analogy to the patches in the ViT (Dosovitskiy et al., 2021) and MAE (He et al., 2022) in the 2D field, existing 3D MAE-based architectures represent a point cloud as a set of point tokens/proxies, making it into a set-to-set translation problem. Attention operators will be applied to depict different spatial local regions to better capture the local geometric and semantic dependence for the tasks of 3D classification, semantic segmentation, etc. We argue that the 3D pose transfer task is different, and the learning focus of the pose source should be made on the extrinsic presentations (i.e., pose). Meanwhile, the traditional 3D spatial-wise attention/correlation operation (Chen et al., 2022; Song et al., 2021) can capture much intrinsic (i.e., shape) information, which is inevitable for tasks like 3D classification, and semantic segmentation. But this is partially inefficient for the task of 3D pose transfer as the detailed geometric information is redundant for learning the pose correlations.

## B  APPENDIX: ALGORITHM FOR THE ADVERSARIAL TRAINING

We show our algorithm for the adversarial training in Algorithm 1.

## C  3D-POSEMAE NETWORK ARCHITECTURE

Our 3D-PoseMAE framework consists of two main parts: the multi-scale masking feature extractor, and the 3D-PoseMAE decoder. We first introduce the network structures of each component, and then give the architectural parameters of the full model.

**Multi-scale Masking Feature Extractor**. The architecture of the feature extractor is presented in Table 7. The feature extractor is used to extract a set of latent embeddings from S-scale representations from the given source pose for further mesh generation with the following decoders. Basically, it works as a normal 3D feature extractor that encodes the input meshes with vertex number $N_1$ into a latent vector with $C$ size. The difference is that the latent vector will be expanded up based on the target meshes along the topology dimension (vertex number $N_2$), resulting in a latent vector with $C \times N_2$. In this way, we align the sizes between source mesh and target mesh as the same and make the learning of the correspondence possible.

**3D-PoseMAE Decoder**. The network architecture of a 3D-PoseMAE decoder is presented in Table 8. Following previous works regarding the 3D pose learning (Peng et al., 2021; Chen et al., 2022), we modified the LayerNorm operation from a classical Transformer architecture into an instance normalization (InsNorm) block inspired by (Wang et al., 2020) presented in Table 9. This is to preserve the geometric structures of the pointcloud while naively using MLP layers will whiten the 3D geometric information. The 3D-PoseMAE decoder is used to generate the posed target mesh with given source pose with full geometric details preserved.

---

**Algorithm 1** Adversarial training with FGM for 3D pose transfer.

---

**Input:** $N$: Total epoch number for training.
  $\lambda$: The threshold to determine a successful attack.
  $\epsilon$: The budget for FGM-based attacks.
  $M_{pose}$: The source pose.
  $M_{id}$: The identity mesh.
  $M_{GT}$: The ground truth mesh.
  $\theta$: The parameter of target model $F$ to attack.
**Output:** $\theta'$: The updated parameter of target model $F$.
  **for** $epoch$ in $N$ **do**
    set $F$ as $eval()$
    $M_{id}.detach(); M_{pose}.detach()$
    $M_{result} = F(M_{pose}, M_{id}; \theta)$
    $\mathcal{L}_{adv} = f_{adv}(M_{result}, M_{GT})$
    $\mathcal{L}_{adv}.backward()$
    $gradient_{adv} = M_{pose}.grad.detach()$
    $\Delta = gradient_{adv} \times \epsilon$
    $M_{adv\_pose} = M_{pose} + \Delta$
    set $F$ as $train()$
    $M_{result} = F(M_{adv\_pose}, M_{id}; \theta)$
    $\mathcal{L}_{rec} = ||M_{result} - M_{GT}||$
    $\mathcal{L}_{rec}.backward()$
  **end for**

---

Table 7: Detailed architectural parameters for the 3D mesh feature extractor. "B" stands for batch size and "N" stands for vertex number. The first parameter of Conv1D is the kernel size, the second is the stride size. "N'" stands for the intermediate pointcloud number. The same as below.

| Index | Inputs | Operation | Output Shape |
|---|---|---|---|
| (1) | - | Input mesh | B×3×N |
| (2) | - | Input mesh | B×3×N/2 |
| (3) | - | Input mesh | B×3×N/4 |
| (4) | (1) | Masking | B×3×N× $\phi$ |
| (5) | (2) | Masking | B×3×N/2× $\phi$ |
| (6) | (3) | Masking | B×3×N/4× $\phi$ |
| (7),(8),(9) | (4),(5),(6) | Conv1D $(1 \times 1, 1)$ | B×64×N' |
| (10),(11),(12) | (7),(8),(9) | Instance Norm, Relu | B×64×N' |
| (13),(14),(15) | (10),(11),(12) | Conv1D $(1 \times 1, 1)$ | B×128×N' |
| (16),(17),(18) | (13),(14),(15) | Instance Norm, Relu | B×128×N' |
| (19),(20),(21) | (16),(17),(18) | Conv1D $(1 \times 1, 1)$ | B×1024×N' |
| (22),(23),(24) | (19),(20),(21) | Instance Norm, Relu | B×1024×N' |
| (25) | (22),(23),(24) | Max pooling and adding | B×1024×1 |
| (26) | (25) | Tiling | B×1024×N |

At last, we present the full model architecture in Table 10. The embedded pose features will be fed into 3D-PoseMAE decoders together with the target mesh and pose mesh will be generated.

## D   DATASET SETTINGS

**Training Sets.** We use SMPL-NPT dataset (Wang et al., 2020) to prepare the training dataset for quantitative evaluation. Note that we only need to train on the SMPL-NPT once, and the model can be generalized and directly conduct the human pose transfer on other testing datasets without further finetuning. SMPL-NPT is a synthesized dataset containing 24,000 body meshes generated via the SMPL model (Bogo et al., 2016) by random sampling in the parameter space. 16 different identities paired with 400 different poses are provided for training. Since the number of paired ground truths will be exponentially huge (24,000*24,000), we randomly select 4,000 training pairs at each epoch during the training.

**Quantitatively Testing Sets.** We use SMPL-NPT dataset (Wang et al., 2020) to prepare the testing set for quantitative evaluation. Following the training stage, 14 new identities (different than that in

Table 8: Detailed architectural parameters for 3D-PoseMAE decoder.

| Index | Inputs | Operation | Output Shape |
|-------|--------|-----------|--------------|
| (1) | - | Identity Embedding | B×C×N |
| (2) | - | Pose Embedding | B×C×N |
| (3) | (1) | conv1d (1 × 1, 1) | B×C×N |
| (4) | (2) | conv1d (1 × 1, 1) | B×C×N |
| (5) | (3) | Reshape | B×N×C |
| (6) | (5)(4) | Batch Matrix Product | B×C×C |
| (7) | (6) | Softmax | B×C×C |
| (8) | (7) | Reshape | B×C×C |
| (9) | (2) | conv1d (1 × 1, 1) | B×C×N |
| (10) | (2)(8) | Batch Matrix Product | B×C×N |
| (11) | (10) | Parameter gamma | B×C×N |
| (12) | (11)(2) | Add | B×C×N |
| (13) | - | Pose Mesh | B×3×N |
| (14) | (12)(13) | SPAdaIN | B×C×N |
| (15) | (14) | conv1d(1 × 1, 1), Relu | B×C×N |
| (16) | (14)(15) | SPAdaIN | B×C×N |
| (17) | (16) | conv1d(1 × 1, 1), Relu | B×C×N |
| (18) | (12)(13) | SPAdaIN | B×C×N |
| (19) | (18) | conv1d(1 × 1, 1), Relu | B×C×N |
| (20) | (17)(19) | Add | B×C×N |

Table 9: Detailed architectural parameters for SPAdaIN block.

| Index | Inputs | Operation | Output Shape |
|-------|--------|-----------|--------------|
| (1) | - | Driving Pose Embedding | B×C×N |
| (2) | (1) | Instance Norm | B×C×N |
| (3) | - | Target Mesh | B×3×N |
| (4) | (3) | Conv1D (1 × 1, 1) | B×C×N |
| (5) | (3) | Conv1D (1 × 1, 1) | B×C×N |
| (6) | (4)(2) | Multiply | B×C×N |
| (7) | (6)(5) | Add | B×C×N |

Table 10: Detailed architectural parameters for the full model.

| Index | Inputs | Operation | Output Shape |
|-------|--------|-----------|--------------|
| (1) | - | Target Mesh | B×3×N |
| (2) | - | Driving Pose Mesh | B×3×N |
| (3) | (2) | Feature Extractor | B×1024×N |
| (4) | (3) | Conv1D (1 × 1, 1) | B×1024×N |
| (5) | (4)(1) | 3D-PoseMAE decoder 1 | B×1024×N |
| (6) | (5) | Conv1D (1 × 1, 1) | B×512×N |
| (7) | (6)(1) | 3D-PoseMAE decoder 2 | B×512×N |
| (8) | (7) | Conv1D (1 × 1, 1) | B×512×N |
| (9) | (8)(1) | 3D-PoseMAE decoder 3 | B×512×N |
| (10) | (9) | Conv1D (1 × 1, 1) | B×256×N |
| (11) | (10)(1) | 3D-PoseMAE decoder 4 | B×256×N |
| (13) | (12) | Conv1D (1 × 1, 1) | B×3×N |
| (14) | (13) | Tanh | B×3×N |

Table 11: Licenses of the assets used in the paper.

| Data | License websites |
|---|---|
| SMPL (Bogo et al., 2016) | https://smpl.is.tue.mpg.de/modellicense.html |
| SMPL-NPT (Wang et al., 2020) | https://github.com/jiashunwang/Neural-Pose-Transfer |
| MANO (Romero et al., 2017) | https://mano.is.tue.mpg.de/license.html |
| DFAUST (Bogo et al., 2017) | https://dfaust.is.tue.mpg.de/license.html |
| FAUST (Bogo et al., 2014) | http://faust.is.tue.mpg.de/data_license |
| Animal (Sumner & Popović, 2004) | https://people.csail.mit.edu/sumner/research/deftransfer/ |

the training set) are paired with 400 poses used in the training set as the "seen" protocol and 200 new poses as "unseen" protocols.

**Qualitatively Testing Sets.** We use the model trained from SMPL-NPT dataset (Wang et al., 2020) to conduct the pose transfer directly on the other human mesh datasets (Bogo et al., 2014; 2017) for quantitative evaluation. Specifically, we chose the meshes from those two datasets which are not strictly an SMPL model. Then we set them as target meshes and drive them with the source poses from the SMPL-NPT (Wang et al., 2020).

**Other Domains.** We also extend the 3D-PoseMAE to other domains. We demonstrate the generalizability of 3D-PoseMAE over the animal domain on the Animal dataset (Sumner & Popović, 2004) and hand domain on the MANO dataset (Romero et al., 2017). The Animal dataset provides correspondences and ground truths of identical motions performed by different animals, such as horse, camel and elephant. Although the vertex number of pose and target meshes are not consistent, i.e., the camel mesh has a different vertex number 21,887 than the horse mesh 8,431, our multi-scale masking encoders can effectively handle it. For hand meshes from MANO dataset, the input and output meshes are all with 778 vertices. Since the topology structures of hands and animals are totally different than human meshes, directly conducting pose transfer with a model trained on human meshes on animals or hands will lead to a degenerated result. Instead, for each domain (i.e., hand or animal), we train the pose transfer on those domain (i.e., hand or animal) as domain-specific learning.

**Raw Scans.** We further extend our method directly on the raw scans from DFAUST dataset (Bogo et al., 2017). To do so, we need to canonicalize the scans from the DFAUST dataset so that the world coordinates can be unified into a norm space as the learnt latent pose space. Specifically, we reset the world coordinate of the target mesh by shifting the vertices to the center and scaling the vertex value into the norm space. Then the pose transfer is conducted to the target mesh. Our method is robust against the global scale and rotation.

**Licenses of the Assets.** The licenses of the assets used in this paper are shown in Table 11. Their licenses are given in the websites.

## E  EXPERIMENTAL IMPLEMENTATION

**Computational setting.** Our algorithm is implemented in PyTorch (Paszke et al., 2019). All the experiments are carried out on a server with four Nvidia Volta V100 GPUs with 32 GB of memory and Intel Xeon processors. We train our networks for 400 epochs with a learning rate of 0.00005 and the Adam optimizer. The weight settings in the paper are $\lambda_{rec}$=1, $\lambda_{edge}$=0.0005. The weight settings directly follow the previous work (Wang et al., 2020). The batch size is fixed as 4 for all the settings. For the first 200 epochs, only pure training with clean samples is conducted to stabilize the model and avoid local minima, where the reconstruction loss and edge loss are used. After 200 epochs, the adversarial training starts with adversarial samples added.

**Design of Adversarial Functions.** As mentioned in the main submission, the goal of constructing the adversarial function is to achieve:

$$F(M_{adv\_pose}, M_{id}; \theta) \neq M_{GT}. \tag{7}$$

Thus it can be converted to maximizing the $||F(x_{adv}; \theta) - M_{GT}||$. However, we cannot naively have:

Table 12: Different attacking methods and their runtimes.

| Methods | PMD↓ ($\times 10^{-4}$) | Runtime/ second per sample |
|---|---|---|
| FGM (Dong et al., 2020) | 237.6 | 0.008 |
| IFGM (Dong et al., 2020) | 244.1 | 0.124 |
| MIFGM (Dong et al., 2020) | 342.3 | 0.131 |
| PGD (Dong et al., 2020) | 242.1 | 0.122 |
| Perturbation (Xiang et al., 2019) | 180.7 | 20.320 |

$$f_{adv} = -||F(M_{adv\_pose}, M_{id}; \theta) - M_{GT}||. \tag{8}$$

Because **the above function cannot be solved by minimizing the loss during the gradient propagation.** Besides, this expression cannot comply with C&W-based attacks. Because for C&W-based attacks, we have:

$$f_{loss} = f_{adv} + f_{dist}, \tag{9}$$

where $f_{dist}$ is the distance between the adversarial samples and original samples. If we implement adversarial function as Equ. 8, it will offset the $f_{dist}$, causing the gradient vanishing problem.

Thus, we naturally think of propose to take the exponential function of $-||F(x_{adv}; \theta) - M_{GT}||$ to convert the maximizing problem into a minimizing problem as below:

$$f_{adv} = e^{-||F(x_{adv}; \theta) - M_{GT}||}, \tag{10}$$

In this way, the adversarial loss term $f_{adv}$ can be positive and decrease when the transfer error gets bigger, fitting our demand. However, in practice, we find that the expression of Equ. 10 cannot provide strong gradient for efficiently generating adversarial samples. Thus we modify the above adversarial function into an even more intuitive way as:

$$f_{adv} = ||F(M_{adv\_pose}, M_{id}; \theta) - M_{GT}||^{-1}. \tag{11}$$

By minimizing the above term, we can push the generated results from the model away from the ground truth mesh, resulting in an attack effect.

**Settings for Adversarial Attacks.** After confirming the adversarial function, we implemented several state-of-the-art 3D adversarial attack methods for a preliminary study, including Fast Gradient Method (FGM) (Dong et al., 2020), Iterative Fast Gradient Method (IFGM) (Dong et al., 2020), Momentum Iterative Fast Gradient Method (MIFGM) (Dong et al., 2020), Projected Gradient Descent (PGD) (Dong et al., 2020) and C&W perturbation (Xiang et al., 2019). For FGM-based methods, the attacking budget are all set as 0.08, the iteration is set as 10, the distance function is L2 norm, the $\mu$ of the MIFGM is 0.1. For C&W perturbation (Xiang et al., 2019), we follow the original setting from the work (Xiang et al., 2019), with the binary search step as 20 and the iteration as 100.

## F APPENDIX: EXPERIMENTS RESULTS

**Different Attacking Methods.** Firstly, we conduct a preliminary evaluation of different attacking methods to choose the best-attacking method for the adversarial training. We implemented different adversarial methods and their runtimes in Table. As shown in Table. 12, we present the pose transfer results using those adversarial samples on a trained model (NPT (Wang et al., 2020)). We can see that the trained NPT model is very sensitive to attacks with large PMD values, meaning the transferred results degenerate. The reason why the Perturbation method shows a smaller PMD is that C&W-based methods have a distance loss term and this term will constrain the adversarial samples to be similar to the original meshes, leading to moderated attacks. Taking both the computational efficiency and adversarial training effectiveness into account, we chose FGM attack (Dong et al., 2020) as the attack type in all the protocols to achieve the adversarial training.

**Different Attacking Budgets.** We present adversarial samples of FGM attacks with different magnitudes with attacking budget scales as 0.08, 0.008, and 0.0008. And as shown in Fig. 6 we present

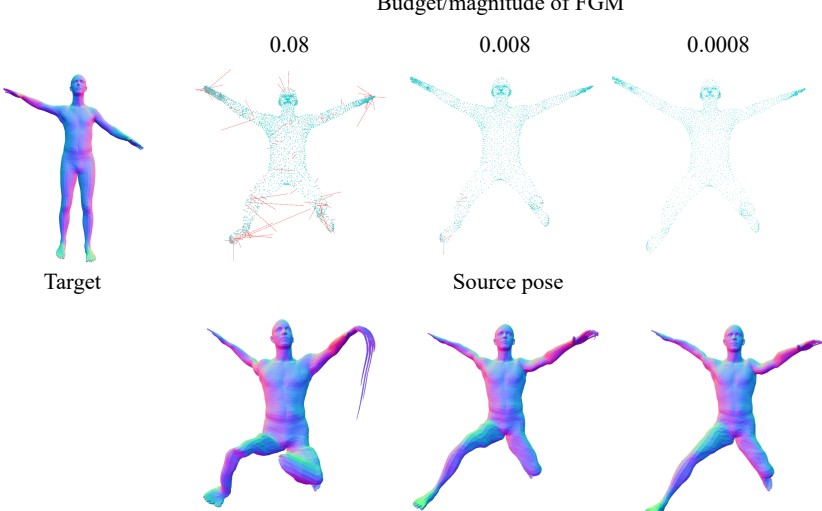

Figure 6: The attacking results of FGM methods on a trained NPT model with different magnitudes. The NPT model is trained with clean samples, so both the pose and appearance of the generated meshes are affected a lot. **We draw the red line to better visualize the perturbed points and we can see that the model is vulnerable to the noisy pose source.**

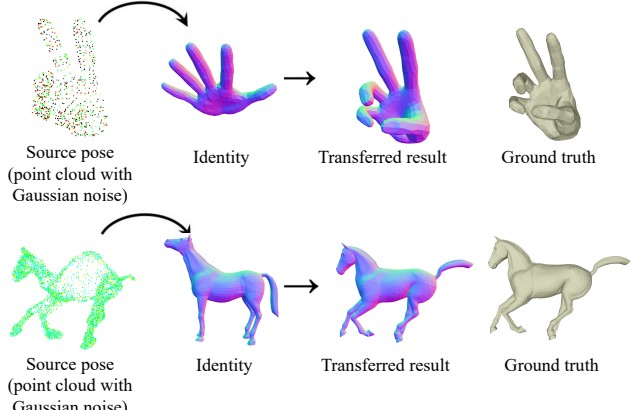

Figure 7: The performance of our method on other domains. We add Gaussian noise to the input point cloud to demonstrate the robustness of the model.

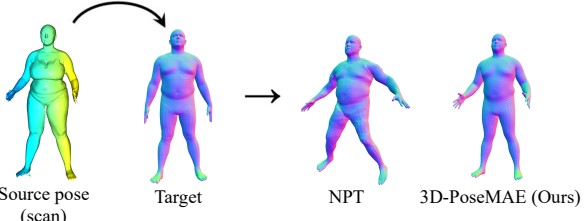

Figure 8: The performance of our method and compared method on raw scans.

the pose transfer results using those adversarial samples on a trained model (NPT (Wang et al., 2020)). We can see that the trained NPT model is very sensitive to the attacks even when the attacking magnitude is small.

**Other Domain.** We use 3D-PoseMAE to achieve pose transfer on meshes from different domains than human meshes, see Fig 7.

**Raw Scans.** We present the pose transfer results of our method and compared method on raw scans from DFAUST dataset as shown in Fig 8.

