# OpenReview forum: "Towards Robust 3D Pose Transfer with Adversarial Learning"
_ICLR.cc/2024/Conference — ICLR 2024 Conference Withdrawn Submission_

### Official Review · Reviewer_MJQ4 · 2023-10-30

**Soundness:** 2 fair
**Presentation:** 3 good
**Contribution:** 2 fair
**Rating:** 5
**Confidence:** 4

**Summary:**

This paper proposes to improve the robustness of the 3D pose transfer task, by modifying architecture designs and introducing adversarial samples during training. For architecture designs, their 3D-PoseMAE consists of multi-scale masking in the encoder and channel attention in the decoder to reduce the reliance on local geometry. For the exploration of adversarial samples, they propose an intuitive way to generate adversarial samples with existing PGD-based methods. Experiments on SMPL-NPT, FAUST and DFAUST show that the proposed method outperforms existing SOTA on noisy raw scans and unseen domains.

**Strengths:**

- They consider a more realistic scenario where the inputs are noisy, which proves to be challenging for previous methods.

- They propose a complete pipeline to improve model robustness by generating adversarial samples and building a more practical model that can deal with them.

**Weaknesses:**

- [Masking] Sec.2.1 claims that existing 3D MAE-based works use multiple subsets to conduct the masking and are not applicable to large-size point clouds. Instead, the proposed multi-scale masking strategy overcomes this challenge. Based on Sec.2.1, the paper highlights their multi-scale masking. However, (Zhang et al., 2022)(add: Yang etal 2023) also use the multi-scale encoder. Therefore, I am not sure why the proposed masking strategy is good for large-size point clouds. For the proposed ratio masking, the description is vague. What are the details of ratio masking? Especially, (add: Yang etal 2023) introduces block/patch/point-wise masking. In this case, it would be better to show the nature of the design to handle large-size point clouds, compared to multi-scale masking methods.


- [Channel-wise Attention] For the design of the decoder, this paper highlights that there are redundant local geometric representations like wrinkles and small tissues. Therefore, the decoder prefers to use channel-wise attention. It is true that there are redundant local geometric representations. I am not sure if totally disregarding local geometric representations with channel-wise attention is reasonable. Based on (add: Hermosilla et al 2019), it seems a multi-level receptive field is a good choice. The channel-wise only attention can be treated as an extreme case. Moreover, after Eq.2, the paper claims *the fine-grained spatial geometric information of the target meshes is preserved by the gradual integration*. Does this mean the decoder tries to learn spatial and geometric information from multi-scale information? This part needs to be introduced in detail.


- [Denoising] Multi-scaling and channel attention are both performing denoising. It would be better to discuss the proposed method in terms of denoising.

- [Adversarial Training and Evaluation] For all the methods w/wo adversarial training, Tab.2 should report the evaluation results on clean data and adversarial samples.

- [Eq.5] Adversarial samples of regression are unlike those of classification. The boundaries among different categories are not applicable for regression tasks. Therefore, Eq.4 does not hold because there is rarely equality in regression tasks. Similarly, Eq.5 is hard to achieve. Because **the direction** and **the magnitude** of adversarial samples are arbitrary, This part should be carefully formulated and discussed. It would be better to discuss adversarial learning (add: Wang et al 2021) or adversarial samples (add: Jain et al 2019) of regression tasks.

- [Motivation] This paper introduces adversarial learning, multi-scaling, and channel attention, which improve the robustness of the model. Those techniques are commonly used for point cloud denoising but seem not to be specially designed for pose transfer. It would be better to highlight the connection between those techniques and pose transfer.

- [Generalization] It would be better to evaluate on unseen datasets to validate the robustness and the generalization, such as SMPL-NPT -> FAUST.



Minor

- Sec 2.2 refers to Eq.7 and Eq.8, which are in the appendix. It would be better to show the important formulas in the main paper.

Reference

- (add: Yang et al 2023) GD-MAE: Generative Decoder for MAE Pre-training on LiDAR Point Clouds. CVPR2023.
- (add: Hermosilla et al 2019) Total Denoising: Unsupervised Learning of 3D Point Cloud Cleaning. ICCV2019.
- (add: Wang et al 2021) When Human Pose Estimation Meets Robustness: Adversarial Algorithms and Benchmarks. CVPR2021.
- (add: Jain et al 2019) On the Robustness of Human Pose Estimation. CVPRW2019.

**Questions:**

See Weaknesses

---

### Official Review · Reviewer_bbpQ · 2023-10-31

**Soundness:** 2 fair
**Presentation:** 2 fair
**Contribution:** 2 fair
**Rating:** 3
**Confidence:** 4

**Summary:**

This paper introduces adversarial learning to the task of 3D pose transfer to improve the generalization capacity. The adversarial samples are generated by minimizing a reversed form of a reconstruction loss between the prediction and the ground truth poses. The authors further propose a channel-wise attention operation based on existing masked autoencoding architecture to learn a more compact and efficient 3D pose representation. Experimental results on both clean and noisy inputs show the better performance of the proposed method compared with existing 3D pose transfer techniques.

**Strengths:**

1. This paper tackles a very practical problem in 3D pose transfer, which is to improve the robustness of 3D pose transfer models such that it can be applied to noisy inputs or raw data directly without tedious preprocessing.
2. The proposed method achieves significant improvements over existing 3D pose transfer methods for both clean and noisy inputs.

**Weaknesses:**

1. The paper is not well structured. Related works are not discussed separately in the paper.  The method part is a bit redundant, especially the section on adversarial training. The motivation, problem definition and also the magnitude of adversarial samples can actually be written in a more precise way.
2. The technical contribution of the proposed method is a bit incremental. The main contributions are the introduction of adversarial training and channel-wise attention, which are adopted from existing works with some modifications.
3. Some related works on 3D pose transfer are lacking.
[a] Keyang Zhou, Bharat Lal Bhatnagar, and Gerard Pons- Moll. Unsupervised shape and pose disentanglement for 3d meshes. In European Conference on Computer Vision (ECCV), 2020.
[b] Jinnan Chen Chen Li Gim Hee Lee. Weakly-supervised 3D Pose Transfer with Keypoints. ICCV 2023
4. Some equations and figures are referred to wrongly. For example, Eq.7 and Eq.8 on the 6th page, Fig.8 on the 9th page.

**Questions:**

Please refer to the weaknesses.

---

### Official Review · Reviewer_Sf1p · 2023-10-31

**Soundness:** 3 good
**Presentation:** 2 fair
**Contribution:** 2 fair
**Rating:** 5
**Confidence:** 3

**Summary:**

A method for 3D pose transfer is proposed in this manuscript. There are three main contibutions. The multi-scale masked encoder for 3D pose reconstruction learning, the 3D-PoseMAE decoder with Channel-wise attention which is designed for efficient pose transfer learning without focusing on redundant local geometric representations (such as wrinkles, small tissues), and a adversarial samples generation method. The proposed method is evaluated on three benchmark datasets, and extensive experiments are conducted for evaluated the performance of it.

**Strengths:**

1. The proposed method achieves state-of-the-art performance.
2. Extensive experiments are conducted to show different aspects of the proposed method.
3. Some pratical attempts, such as the muti-scale masking strategy, are introduced in this paper.

**Weaknesses:**

1. The novelty of the proposed method is limitted. Similar ideas, such as multi-scale learning, channel-wise attention, and the objectives for adversarial sample generation in Eq.5 can also be seen in other papers of similar tasks [1,2]. The proposed ideas are good attempts for application, but are not novel enough for academic publication.
2. For the 3D pose transfer task, to generate 3D pose that is similar to the ground truth is one of the goal. Anohter goal should be generating a physically convincing 3D poses. For example, interpenetration usually happens when the input 3D pose shape is skinny and the shape of the output pose is fat. The transferred pose should keep the motion semantics of the input pose and also avoids introducing flaws such as interpenetration [3]. The flaw avoidance part is missed by this manuscript.
3. Some minors. For example in page 6, Eq.4 and Eq.5 are wrongly marked as Eq. 7 and Eq. 8.

[1] Villegas, R., Yang, J., Ceylan, D., & Lee, H. (2018). Neural kinematic networks for unsupervised motion retargetting. In Proceedings of the IEEE conference on computer vision and pattern recognition (pp. 8639-8648)
[2] Rempe, D., Birdal, T., Hertzmann, A., Yang, J., Sridhar, S., & Guibas, L. J. (2021). Humor: 3d human motion model for robust pose estimation. In Proceedings of the IEEE/CVF international conference on computer vision (pp. 11488-11499).
[3] Zhang, J., Weng, J., Kang, D., Zhao, F., Huang, S., Zhe, X., ... & Tu, Z. (2023). Skinned Motion Retargeting with Residual Perception of Motion Semantics & Geometry. In Proceedings of the IEEE/CVF Conference on Computer Vision and Pattern Recognition (pp. 13864-13872).

**Questions:**

1. As the proposed method is evalauted on the digital human body dataset, the intrinsic structure of human body is well-learned. I am wondering whether the proposed method could also achieve good performace on cartoon mesh, for example the mixamo dataset. As for the cartoon character, the transfer of mesh details such hair, cloth are also important.
2. What about trained on human body and test on cartoon data?

---

### Official Review · Reviewer_qTTF · 2023-11-03

**Soundness:** 2 fair
**Presentation:** 2 fair
**Contribution:** 2 fair
**Rating:** 3
**Confidence:** 3

**Summary:**

This work aims to improve the model's robustness to the input in 3D pose transfer task. An adversarial sample based training pipeline is introduced, together with a novel 3D pose masked autoencoder. The generating of adversarial examples and multi-scale masking are introduced to make the 3D pose transfer robust to the noise in input poses and even further able to be extended to real-world data like raw point clouds/scans. Experimental results demonstrate better quality of the transferred meshes and model generalizability.

**Strengths:**

The multi-scale masked 3D encoder could save computation as the aggregated feature is calculated.

The 3D poseMAE decoder with channel-wise attention is different from the spatial attention in the GC-Transformer. But the contribution is minor.

Promising experimental results are demonstrated.

**Weaknesses:**

The masking in multi-scale masked 3D encoder is similar to random sampling the point cloud for encoding, but in the concept of mask auto-encoder. The idea of masking does not share much with the idea of He et al, 2022, which applies heavy masking in the input and predicts the masked content.

The idea of adversarial training for 3D pose transfer does not make sense. The adversarial training in computer vision tasks mostly has different input and output spaces, e.g. image space and semantic space for classification task. The adversarial training then aims to make the output space robust to small perturbation in the input space. However, in the 3D pose transfer task, the input and output spaces are the same. Any perturbation on the input mesh should be reflected in the output mesh.

The overall process of the adversarial training is not clear. As I understand, there should be a function for the generation of adversarial samples, which I suppose to be equation 5. There should be another loss function to train the model with both the original input samples and the adversarial samples. However, the design of this loss function is missing. The mentioned equation is not the function that will be used.

The generation of adversarial samples seems to be very inefficient.

The main content is expected to be self-contained. However, Eq. 8 is mentioned on page 6, but described in the appendix.
The texts in Figure 3 are too small to recognize, especially when printed on paper.

**Questions:**

Is there any difference between the masking in multiscale masked 3D encoder with point cloud random sampling? Are other sampling methods, like FPS based sampling considered in applying the masking?

What is the ratio of easy samples that are filtered out by SOR?

Can you provide some examples of the adversarial samples? How to set the magnitude of the perturbation?

---

### Official Review · Reviewer_yQax · 2023-11-05

**Soundness:** 2 fair
**Presentation:** 2 fair
**Contribution:** 2 fair
**Rating:** 5
**Confidence:** 4

**Summary:**

In this paper, the authors propose adversarial training for the task of 3D pose transfer using a novel architecture based on multi-scale MAE and channel-wise attention in the decoder. In particular, they rely on dynamic adversarial samples-based data augmentation (computed on the fly for every batch) to conduct adversarial training. The authors empirically validate the benefit of their proposed approach on three datasets and show that their method outperforms the considered baselines.

**Strengths:**

- The presentation of the paper is good.
- Individual components in the proposed architecture are detailed sufficiently.

**Weaknesses:**

- Limited novelty, as both components, MAE and adversarial training, are well-known in the literature. In particular, the adversarial training performed in the current setting is well-known in image classification literature where the adversarial samples are generated for every batch.

- The core novel additions in the current paper are multi-scale MAE and channel-wise attention in the decoding block.  Although, these provide benefit in performance, frameworks such as Multi-scale MAE has already been proposed in literature [a].

- Besides, evaluation against adversarial attacks is incomplete (only considering white-box attacks).

- There is no proper related work section in the paper. For instance, it is difficult for the reader to understand the considered baselines such as NPT, GC-transformer, and the critical differences. Table 4 is helpful, but the authors should consider adding a section after the introduction. I suggest adding a related work section on 3D pose transfer methods, 3D pose data augmentation methods, and 3D adversarial training and defenses. In the current version, the related work is quite sparse.


References:

a. Zhang, R., Guo, Z., Gao, P., Fang, R., Zhao, B., Wang, D., ... & Li, H. (2022). Point-m2ae: multi-scale masked autoencoders for hierarchical point cloud pre-training. Advances in neural information processing systems, 35, 27061-27074.

**Questions:**

- Please clarify the attack performed in Table 2 for evaluation against adversarial samples. Is this a FGM or PGD attack?

- In Table 2, what is the performance of adversarial training on the clean samples? Do you observe a standard accuracy-robustness tradeoff in this task?

- My other concern is that FGSM+AT is shown to be vulnerable to PGD attacks and also leads to catastrophic overfitting (CO) during training [a]. As such, several improvements [b,c,d, e] to FGSM+AT have been studied to enhance performance against adversarial attacks. In my view, the authors should carefully assess the robustness of FGSM+AT against strong attacks and also clarify categorically if they observed CO during training

- What is the performance on clean and adversarial samples with different attacking budgets in FGM training? How to set this attack budget to achieve the proper balance between performance on clean and adversarial samples?


- Authors should conduct an ablation with more advanced data-augmentation strategies to understand the true benefit from adversarial training (AT). Maybe adding noise to the input during training can also be considered as a way to decrease overfitting. There are also advanced pose augmentation methods such as [f] and maybe even more advanced ones. The authors should benchmark against these augmentation strategies and show the benefits of AT compared to them.

Overall, the paper lacks thorough evaluation with strong data augmentation schemes with training on the clean samples, nor does it show that the proposed adversarial training (AT) is robust to the strongest of the attacks.

a. Rice, L., Wong, E., & Kolter, Z. (2020, November). Overfitting in adversarially robust deep learning. In International Conference on Machine Learning (pp. 8093-8104). PMLR.

b. Andriushchenko, M., & Flammarion, N. (2020). Understanding and improving fast adversarial training. Advances in Neural Information Processing Systems, 33, 16048-16059.

c. Vivek, B. S., & Babu, R. V. (2020, June). Single-step adversarial training with dropout scheduling. In 2020 IEEE/CVF Conference on Computer Vision and Pattern Recognition (CVPR) (pp. 947-956). IEEE.

d. Li, B., Wang, S., Jana, S., & Carin, L. (2020). Towards understanding fast adversarial training. arXiv preprint arXiv:2006.03089.

e. Kim, H., Lee, W., & Lee, J. (2021, May). Understanding catastrophic overfitting in single-step adversarial training. In Proceedings of the AAAI Conference on Artificial Intelligence (Vol. 35, No. 9, pp. 8119-8127).

f. Gong, K., Zhang, J., & Feng, J. (2021). Poseaug: A differentiable pose augmentation framework for 3d human pose estimation. In Proceedings of the IEEE/CVF conference on computer vision and pattern recognition (pp. 8575-8584).